# Iron Chelation in Patients with Myelodysplastic Syndromes and Myeloproliferative Neoplasms—Real-World Data from the German Noninterventional Study EXCALIBUR

**DOI:** 10.3390/jcm12206569

**Published:** 2023-10-17

**Authors:** Felicitas Schulz, Ulrich Hauch, Sandra Ketzler-Henkel, Eyck von der Heyde, Michael Koenigsmann, Michael Lauseker, Nora Schulte, Ulrich Germing

**Affiliations:** 1Department for Hematology, Oncology and Clinical Immunology, University Hospital Duesseldorf, 40225 Düsseldorf, Germany; germing@med.uni-duesseldorf.de; 2Practice for Hematology and Oncology, 99084 Erfurt, Germany; 3Practice for Hematology and Oncology, 44309 Dortmund, Germany; 4Oncologic Center at Raschplatz, 30161 Hannover, Germany; vdh75@gmx.de; 5MediProjekt GbR, 30171 Hannover, Germany; koenigsmann@onkologie-hannover.de; 6Institut für Medizinische Informationsverarbeitung Biometrie und Epidemiologie (IBE), Fakultät für Medizin, Ludwig-Maximilians Universität München, 81377 Munich, Germany; lauseker@ibe.med.uni-muenchen.de; 7Novartis Pharma GmbH, 90429 Nuremberg, Germany; nora.schulte@gmx.net

**Keywords:** iron chelation, iron overload, myelodysplastic syndromes, MDS, myeloproliferative neoplasms, MPN, ferritin, hematologic improvement

## Abstract

Myelodysplastic syndromes and myeloproliferative neoplasms both represent hematologic diseases associated with bone marrow failure often resulting in anemia. For those patients, transfusion of red blood cell (RBC) units is essential but results in iron overload (IOL) that may affect various organ functions. Therefore, iron chelation therapy plays a major role in anemic patients, not only because it reduces IOL, but also because it may improve hematopoietic function by increasing hemoglobin or diminishing the requirement for RBC transfusions. To assess the utility, efficacy, and safety of the different iron chelation medications approved in Germany, as well as to examine the effect of chelation on hematopoietic insufficiency, a prospective, multicenter, noninterventional study named EXCALIBUR was designed. In total, 502 patients from 106 German hospitals and medical practices were enrolled. A large proportion of patients switched from a deferasirox dispersible tablet to a deferasirox-film-coated tablet, mainly because of more convenient application, which was reflected in the treatment satisfaction questionnaire for medication scores. Iron chelation was effective in lowering serum ferritin levels, with the observed adverse drug reactions being in line with the known safety profile. Hematologic response occurred in a few patients, comparable to other studies that examined hematologic improvement in patients with MDS.

## 1. Introduction

Myelodysplastic syndromes (MDS) and myeloproliferative neoplasms (MPN) both represent hematologic diseases associated with bone marrow failure eventually resulting in cytopenia. Clinically, anemia is one of the leading problems in MDS and, in part, MPN patients, and transfusion of red blood cell (RBC) units is essential but results in iron overload (IOL). Physiologically, the human body requires a certain amount of iron for important cellular processes such as energy acquisition or oxygen transport [1]. Unfortunately, no physiological mechanism exists to dispose of excess iron. Iron overload increases the level of oxidative stress via an increase in chronic exposure to non-transferrin-bound iron, leading to damage of macromolecules like DNA, proteins, or lipids [2]. In MDS patients, IOL is already present before patients become transfusion dependent because ineffective erythropoiesis suppresses the production of hepcidin in the liver, resulting in unregulated iron uptake in the intestine [3,4]. Nonetheless, chronic transfusion therapy is the most important cause of iron overload in patients with MDS [5]. Each unit of red blood cell concentrates leads to the substitution of 200 mg of iron. As an example, this causes an intake of about 20 g of iron within 2 years in patients receiving four RBC units per month [5]. As anemia leads to a significantly increased risk of death from cardiac complications in patients with MDS, transfusion therapy is vital [5]. Because transfusion therapy is accompanied by higher iron levels and IOL means an additional cardiac risk factor, this represents the vicious circle of every anemic hematologic patient with MDS or MPN and, in general, for any patient with chronic transfusion need, e.g., because of thalassemia. However, not only is cardiac iron overload detectable after transfusion of multiple RBC units, but other organs are affected from iron overload as well, such as the liver and the endocrine glands. Thus, IOL may cause heart failure and arrhythmias, as well as liver fibrosis and cirrhosis, diabetes mellitus, and hypothyroidism [1]. Furthermore, IOL might result in genomic instability, thereby possibly encouraging clonal evolution toward leukemia [5]. Therefore, iron chelation therapy (ICT) plays a major role in anemic patients, not only because it reduces IOL, but also because it may improve hematopoietic function by increasing hemoglobin or diminishing the requirement for RBC transfusions [2]. There are different iron chelators available in Germany, concomitant with different ways of application. For several decades, only intravenous or subcutaneous deferoxamin (Desferal^®^) was available, followed by deferasirox (Exjade^®^) as a dispersible tablet (DT) with the need to disperse into a suspension prior to consumption. During the course of the study discussed here, deferasirox became available as a film-coated tablet (FCT) formulation that just needs to be swallowed [6], which is meanwhile the only formulation of deferasirox available in Germany. For patients with thalassemia major, the iron chelator deferiprone (Ferriprox^®^) is available as an FCT or a solution for oral administration, as a monotherapy, or in combination with another iron chelator. Deferiprone is used in exceptional cases (e.g., if there is a contraindication for deferasirox) in other relevant indications, such as MDS, too. All of these drugs may lead to a decrease in serum ferritin levels via elevated elimination of iron through feces while showing a favorable profile of side effects. The exact percentage distribution of the choice of treatment in Germany is unknown. Quantification of IOL in clinical daily practice is measured via patients’ serum ferritin, and treatment is mostly initiated when serum ferritin levels are higher than 1000 µg/L and repetitive RBC transfusions are necessary. A dose-dependent impact of IOL on overall survival of patients has been demonstrated for this serum ferritin threshold, observing a 30% higher risk of death for each 500 µg/L increase in serum ferritin above 1000 µg/L [7]. Furthermore, a large retrospective analysis showed that IOL was a significant prognostic factor not only for OS, but also for leukemia-free survival in MDS [8]. To assess the utility, efficacy, and safety of the iron chelation medication approved in Germany, a prospective, multicenter, noninterventional study producing real-world data named EXCALIBUR was designed (NCT05440487; CICL670ADE14; funded by Novartis, Nuernberg, Germany). Additionally, the effect of chelation on hematopoietic insufficiency was examined.

## 2. Materials and Methods

EXCALIBUR was a prospective, multicenter, noninterventional study assessing iron chelation therapy in patients with chronic iron overload. Male and female adult patients suffering from chronic iron overload who never received an iron chelator, who received an iron chelation therapy for less than 6 months, or who interrupted an iron chelation therapy for longer than 6 months were included. In total, 502 patients from 106 German hospitals and medical practices for hematology and oncology, who met the inclusion criteria and signed the written informed consent, were enrolled. The entire duration of the study was approximately 6 years. Treatment with iron chelators did not follow a predefined protocol, but was administered according to routine medical practice. The observation period was 24 months for patients without a change of iron chelator and was extended by 24 months if the iron chelator was changed. Follow-up visits were documented after 1, 3, 6, 9, 12, and 18 months, with a final visit after 24 months or at the end of the observation phase, whichever occurred first. In- and exclusion criteria were based on the summary of product characteristics of the respective iron chelator, i.e., deferasirox DT, deferasirox FCT, and deferoxamine. Patients’ general satisfaction with all approved iron chelators in everyday life was assessed using the treatment satisfaction questionnaire for medication (TSQM-14) after approx. 1 and 3 months. Data were collected via an eCRF by the treating physician or authorized personnel. Statistical analyses were performed using the software package SAS release 9.4. Continuous data were described by the number of patients in the respective population, non-missing and missing values, mean, standard deviation, median, and interquartile range. Categorical data including categories of continuous data were presented in frequency tables containing absolute and relative frequencies. For the hematological response analyses, patients’ cumulative incidences of the respective response were estimated considering death without response as a competing risk. Analysis of hematological response regarding erythroid, platelet, and neutrophil response followed the criteria of the international working group (IWG) [9]. Those patients who had already fulfilled the response criteria at baseline, or who were not evaluable at that time, were excluded from hematological response analyses. For the analyses of changes in hematological parameters, mixed linear regression models were estimated.

## 3. Results

Overall, 502 patients were enrolled in the study database, with 418 patients in the safety analysis set (SAF), 403 patients in the full analysis set (FAS), and 266 patients in the hematological response analysis set (HRAS). Reasons for exclusion from the different analysis sets are shown in Figure 1. Of the 106 registered study sites, 101 sites had at least one patient included in the SAF, and 98 sites had at least one patient included in the FAS. Median age of patients at baseline was 75 years (ranging from 24 to 92) with a higher proportion of males (59.6%). Patient demographics and disease characteristics are shown in Table 1. The most common hematologic diagnosis was MDS (61.0%), followed by MPN (16.1%). Subtypes of MDS and MPN diagnoses are shown in Table 2, with MDS with multilineage dysplasia (MDS-MLD, 19.5%), MDS with excess of blasts I (MDS-EB I, 13%), and MDS with single lineage dysplasia and ring sideroblasts (MDS-SLD-RS, 12.2%) being the most frequent subtypes of MDS, and primary myelofibrosis for MPN (63.1% of MPN patients). The median time from first diagnosis to current iron chelation therapy was 21.6 months (with an interquartile range of 8.6 to 50 months), and only 5.2% of patients had iron chelation therapy prior to study inclusion. The vast majority of patients had concomitant diseases like hypertension (50.4%), coronary artery disease (11.9%), or diabetes (10.2%), as well as concomitant medication (94.0%). The median time from primary diagnosis to receiving transfusions was 2.8 months (interquartile range 8.6 to 50 months), and the median time from start of transfusions to start of current iron chelation therapy was 12.1 months. More than 95% of patients had received at least one red blood cell transfusion prior to study entry, while most of the patients had received less than 20 erythrocyte concentrates (47.3%) prior to study entry.

### 3.1. Use and Switching of Iron Chelators

In total, 267 patients started with deferasirox FCT (66.3%), while 111 patients received deferasirox DT (27.5%), and 25 patients received deferoxamine (6.2%). During the entire observation period, 310 patients were treated with deferasirox FCT (76.9%), 116 patients with deferasirox DT (28.8%), and 30 patients with deferoxamine (7.4%). The mean time to premature discontinuation of treatment was approximately 9 months, and 77.9% of patients discontinued prematurely. This was similar for patients who were last treated with deferasirox FCT and deferoxamine. Notably, 93.2% of the 74 patients who were last treated with deferasirox DT discontinued their treatment prematurely. The main reasons for premature discontinuation were AEs (29.9%) and death (28.3%) with a median time to premature discontinuation of 200 days. The reason for treatment initiation was a high serum ferritin value of more than 1000 ng/mL in 90.8% of cases, followed by transfusion of more than 20 red blood cell units in 30.3% of patients. The frequencies of treatment changes are shown in Table 3. Overall, 11.9% of patients had treatment changed once, most of them from deferasirox DT to deferasirox FCT. Only four patients changed iron chelator twice. The most commonly reported reason for treatment change was intricate application (48.8%). After the end of the study, all 89 patients without premature discontinuation continued the same treatment. The mean ± SD initial daily dose of the start treatment deferasirox DT was 11.7 ± 6.29 mg/kg, while for deferasirox FCT it was 13.1 ± 6.77 mg/kg. The initial daily dose of deferoxamine was 25.6 ± 10.22 mg/kg. The mean ± SD difference of initial vs. last dose of start treatment was +1.3 ± 4.13 mg/kg for deferasirox DT, +0.7 ± 5.77 mg/kg for deferasirox FCT, and +6.2 ± 15.36 mg/kg for deferoxamine. The mean ± SD number of dosage adjustments per patient was 0.8 ± 1.38 (n = 116) for deferasirox DT, 1.1 ± 1.41 (n = 310) for deferasirox FCT, and 1.2 ± 1.76 (n = 30) for deferoxamine.

### 3.2. General Satisfaction with Iron Chelation Treatment

The general satisfaction with iron chelation treatment was measured with the standardized patient questionnaire TSQM-14, which consisted of 14 questions assigned to 4 subscales. The questionnaire was answered at month 1 and month 3 of the initial treatment and again at month 1 and month 3 after treatment change, if applicable. The median (IQR) overall satisfaction score was 68.1 (50.0–77.8) points (n = 224) at month 1 and 63.9 (50.0–77.8) points (n = 181) at month 3 of the initial treatment. After treatment change, the score reached 73.6 (52.8–87.5) points (n = 16) at month 1 and 80.6 (47.2–91.7) points (n = 18) at month 3. The differences show a tendency toward higher satisfaction after treatment change and even higher satisfaction after two months following the change, but this was statistically not relevant.

### 3.3. Safety

The vast majority of patients experienced AEs (92.6%) and around two-thirds of patients experienced serious AEs (SAEs). For around 50% of patients, non-serious events were assessed to have a suspected causal relationship with the respective treatment; thus, they were classified as adverse drug reactions (ADRs), and 20.6% of patients had serious ADRs (SADRs) (Table 4). Overall, the most common non-serious AEs (nsAEs) were fatigue (12.1% of patients with nsAEs), dizziness, and nausea (10.2% each). The most common serious AEs (SAEs) were pneumonia (12.6% of patients with SAEs) and general physical health deterioration (9.3%). The most common non-serious adverse drug reactions (nsADRs) were diarrhea (25.9% of patients with nsADRs), increase in blood creatinine (13.2%), and nausea (10.0%). The most common serious adverse drug reactions (SADRs) were renal failure (11.6% of patients with SADRs) and an increase in blood creatinine (8.1%). Detailed listings of the most common patient-based (S)AEs and (S)ADRs are shown in Table 5.

### 3.4. Effectiveness and Transfusion Dependence

The median serum ferritin value decreased notably throughout treatment, with 1802.50 µg/L at baseline and 1240.50 µg/L at month 24. The median change from baseline at month 24 was −458.0 µg/L; the change from baseline throughout treatment is shown in Figure 2. At baseline, patients had received a median of 4 (IQR: 3-8) erythrocyte concentrates within the last 8 weeks. At all subsequent visits throughout the initial treatment, patients had received a median of 6 erythrocyte concentrates within the last 8 weeks.

### 3.5. Hematological Response

Changes in blood parameters were estimated based on mixed linear models using the time as independent variable and the patient as random. Cumulative incidence at 24 months for hemoglobin response was 15.2% (95% CI: 9.6–22.0%) for MDS patients and 9.3% (95% CI: 2.3–22.4%) for MPN patients with increasing mean hemoglobin values from 8.3 (95% CI: 8.2–8.5) to 8.9 (95% CI: 8.5–9.2) g/dL for MDS patients and from 8.3 (95% CI: 8.0–8.7) to 8.9 (95% CI: 8.4–9.3) g/dL for MPN patients. Cumulative incidence at 24 months for transfusion response was 16.3% (95% CI: 9.4–24.8%) for MDS patients and 23.1% (95% CI: 6.4–45.9%) for MPN patients. Mean platelet values decreased from 145.4 (95% CI: 127–166) to 103.4 (95% CI: 85–125) × 10^3^/µL for MDS patients and from 112.7 (95% CI: 84–151) to 80.2 (95% CI: 58–111) × 10^3^/µL for MPN patients. Nonetheless, 24-month cumulative incidences of platelet response were 18.9% (95% CI: 9.1–31.5%) for MDS patients and 31.2% (95% CI: 7.6–59.1%) for MPN patients. Cumulative incidence at 24 months for neutrophil response was 30.0% (95% CI: 11.4–51.3%) for MDS patients, while none was observed in the MPN patients, with mean neutrophil values that increased from 3.0 (95% CI: 2.8–3.3) to 3.4 (95% CI: 2.9–3.9) × 10^3^/µL for MDS patients, and from 5.3 (95% CI: 4.3–6.5) to 5.9 (95% CI: 4.7–7.5) × 10^3^/µL for MPN patients (Table 6). Cumulative incidences of hematological responses, as well as cumulative incidence curves regarding time to hematological response, are shown in Table 6 and Figure 3. We observed 30 responses regarding hemoglobin, of which 6 were lost during the observation time. Eight out of 19 patients who showed transfusion response lost that response during the course of the study, while 3 out of 13 patients lost their response regarding transfusions. All patients who showed neutrophil response maintained that response until the end of the observational period.

## 4. Discussion

EXCALIBUR was a prospective, noninterventional study that was performed under conditions of clinical daily practice in Germany, thereby allowing the enrollment of a heterogeneous patient population with regard to demographic and disease characteristics. The inclusion of patients with various diagnoses treated with three different iron chelators represented a realistic population, and the observational design of the study allowed the collection of real-life data without influencing the physicians’ treatment decisions. To minimize possible study site effects, a large number of different hospitals and medical practices were enrolled, depicting a geographically representative selection of German sites. As the study was only conducted in Germany, the generalizability of the results for other countries may be limited. Due to the observational aspect of the study, there are associated limitations like the lack of blinding and randomization, as well as a relevant amount of missing or inconsistent data. However, the overall data quality reflects a typical MDS population, as well as a real-life treatment situation, and provides information regarding clinical practice and patient behavior that randomized clinical trials do not obtain.

EXCALIBUR evaluated the use and application of all approved iron chelators with respect to safety, tolerability and patient satisfaction, including treatment switches, as well as the effectiveness of iron chelation therapy and the hematological response in MDS/MPN patients. As the treating physician decided on the prescription of medication and inclusion of the patient in this NIS, there was a potential aspect of influencing the patients’ decisions and course of treatment, herewith introducing bias. To account for the effect of premature withdrawals and treatment changes, the data for all patients at the last completed visit were summarized in the form of a “last visit” in the analyses. The patient characteristics within the study were in line with previous studies conducted in Germany—for example, the 2-year prospective observational study that aimed at describing the routine use of DFX in patients with hematological malignancies by Nolte et al. [10].

One main objective of our study was to assess the use of approved iron chelators in Germany under real-life conditions. Deferasirox DT was released in Germany in 2006 as a more convenient alternative to deferoxamine, which had to be administered parenterally. However, some subjective burden remained as deferasirox DT had to be dispersed in a relatively time-consuming procedure, taken on an empty stomach, and was associated with relevant gastrointestinal side effects [11,12]. Therefore, the film-coated tablet formulation was introduced in 2016. Within EXCALIBUR, Deferasirox FCT was prescribed as initial treatment for the majority of patients, followed by deferasirox DT and deferoxamine. Deferiprone, the fourth approved iron chelator in Germany, is indicated for the treatment of iron overload in patients with thalassemia major when current chelation therapy is contraindicated or inadequate. Due to this specific indication, only two patients had deferiprone as starting treatment and were therefore excluded from further analyses. Most patients with a documented treatment change switched from deferasirox DT to deferasirox FCT (83.3%). The main reason for this treatment change was the intricate application of deferasirox DT. Later on, deferasirox DT was no longer available, thus requiring all patients to switch.

With the TSQM-14 questionnaire, convenience was assessed with a slightly lower score by patients with deferasirox DT as starting treatment compared to the overall population. These results indicated that patients indeed preferred deferasirox FCT over deferasirox DT and deferoxamine because of the changed formulation, which is in line with the ECLIPSE study that could show consistently greater adherence and higher satisfaction for deferasirox FCT compared to deferasirox DT [6]. A patient’s satisfaction measured by the TSQM-14 questionnaire ranged between 59 and 86 points for the different subscales and start treatment groups, with the maximum score being 100 points. The overall satisfaction ranged between 61 and 72 points. There was a slight tendency toward decreasing scores from month 1 to month 3, and toward increasing scores after treatment change. Effectiveness and side effects were assessed with slightly higher scores for deferoxamine compared to the overall population; however, smaller patient numbers should be considered.

The ADRs observed in this study were in line with the known safety profiles of deferasirox and deferoxamine [13,14]. Overall, the most common nsADRs were diarrhea (25.9% of patients with nsADRs) and an increase in blood creatinine (13.2%). The most common SADRs were renal failure (11.6% of patients with SADRs) and, again, an increase in blood creatinine (8.1%). This was similar for patients treated with deferasirox DT and deferasirox FCT and, in general, matches the results of the TELESTO trial, a randomized trial that assessed iron chelation in transfusion-dependent patients with low- to intermediate-1-risk myelodysplastic syndromes [15].

The main reason for initiation of iron chelation was a serum ferritin value of >1000 ng/mL. To assess the effectiveness of iron chelation therapy, serum ferritin values were repeatedly measured throughout the study. The median serum ferritin value decreased from 1802.50 µg/L at baseline to 1240.50 µg/L at month 24; the median change from baseline at month 24 was −458.0 µg/L. Persistence of elevated serum ferritin levels is most likely due to the continuing transfusion dependence of the patient cohort. In the phase IIIb EPIC study, patients with MDS showed comparatively larger decreases between −115 µg/L and −976 µg/L after 1 year of treatment with deferasirox [11]. However, the comparability is limited by different study designs and underlying patient populations. A 3-year, prospective, multicenter trial that assessed the safety and efficacy of deferasirox in low- or intermediate-1-risk MDS by List et al. documented a median serum ferritin decrease of 36.7% in patients who completed 2 years, which is approximately in line with our results with EXCALIBUR [16]. Due to the fact that there is no definitive trigger to start iron chelation therapy, but treatment initiation happens at the discretion of the treating physician, there might be a relevant variation between different centers, hereby introducing bias. Looking at a consensus statement on iron overload in myelodysplastic syndromes, iron chelation should be initiated when ferritin levels are >1000 ng/mL [17,18,19], which might be the widespread reason for beginning iron chelation. In patients with a high transfusion burden and continuing need for transfusion of red blood cell units, earlier treatment initiation is reasonable as well. Regarding the patient cohort of EXCALIBUR, the main reasons for iron chelation treatment were a serum ferritin value of >1000 ng/mL and RBC transfusion of >20 units. In line with the reason for starting treatment of iron overload, a ferritin level < 1000 ng/mL might be a therapeutic goal to prevent iron accumulation in various organs.

For patients with MDS and MPN, the hematological response was determined according to IWG criteria. In this competing-risks setting, death without prior response usually had a higher probability than the achievement of a response. At 24 months, the probability of hemoglobin response was 15.2% for MDS patients and 9.3% for MPN patients. After the same time, the probability of transfusion response was 16.3% for MDS and 23.1% for MPN patients. At 24 months, we estimated that 18.9% of MDS patients and 31.2% of MPN patients reached platelet response, and for 30.0% of MDS patients but for none of the MPN patients, a neutrophil response could be documented, which might be due to the small number of MPN patients being evaluable for neutrophil response. When performing literature research, there are more than 10 studies assessing the rates of hematologic improvement in patients with MDS receiving iron chelation therapy between 2012 and 2019 [20]. Compared to the post hoc analysis of the EPIC trial, for example, in which 21.5% of patients responded regarding hemoglobin, 13.0% of patients responded regarding platelets, and 22% when it came to neutrophils [21], results were similar to ours except for hemoglobin response. Compared to the results of the GIMEMA MDS0306 trial, a prospective, open-label, single-arm, multicenter trial of transfusion-dependent patients with IPSS low- or intermediate-1-risk MDS assessing the safety and efficacy of deferasirox, we could document a higher amount of neutrophil response (3% within GIMEMA) but almost the same incidence of erythroid and platelet response (11% and 15%, respectively) [22]. Within a ‘real-world’ report from two regional Italian registries, Maurillo et al. assessed deferasirox chelation therapy in patients with transfusion-dependent MDS with an erythroid response of 17.6% (with a 7.1% rate of RBC transfusion independence), a neutrophil response of 7.1%, and a platelet response of 5.9% [23]. In summary, compared to other pro- and retrospective analyses, the results of EXCALIBUR regarding hematologic improvement/response seemed to be representative. While a matched-pair analysis from the Düsseldorf MDS registry showed improved survival in patients receiving iron chelation therapy [24], we could not give evidence concerning the survival of our study population because of the observation period of only six years.

## 5. Conclusions

EXCALIBUR provided valuable insights into the utilization, effectiveness, safety, and patient satisfaction of iron chelation therapy in clinical daily practice, while additionally documenting change of iron chelator medication over time. The observational design of the study allowed the enrollment of a heterogeneous patient population with regard to demographic and disease characteristics, thereby collecting real-life data without influencing the physicians’ treatment decisions. The prospective documentation in a broad patient population with various diagnoses treated with three different iron chelators led to a high representativeness of a realistic patient population. A large proportion of patients switched from deferasirox DT to deferasirox FCT, mainly because of more convenient application, and this was also reflected in the TSQM-14 scores. Iron chelation was effective in lowering serum ferritin levels, with the observed ADRs being in line with the known safety profile. Hematologic response occurred in a small number of patients, comparable to other studies that examined hematologic improvement in patients with MDS.

## Figures and Tables

**Figure 1 jcm-12-06569-f001:**
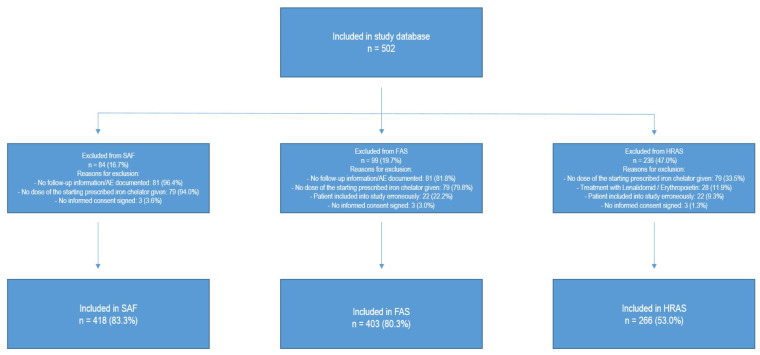
Analysis sets. Abbreviations: FAS = full analysis set, HRAS = hematological response analysis set, SAF = safety analysis set.

**Figure 2 jcm-12-06569-f002:**
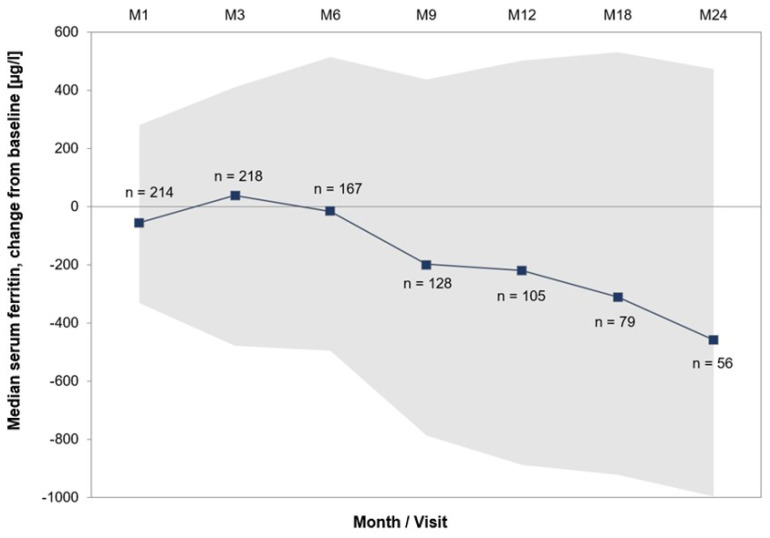
Course of serum ferritin values by study months (M1-M24). Gray area shows IQR. Abbreviations: IQR: interquartile range, M: month, n: number of patients with observation.

**Figure 3 jcm-12-06569-f003:**
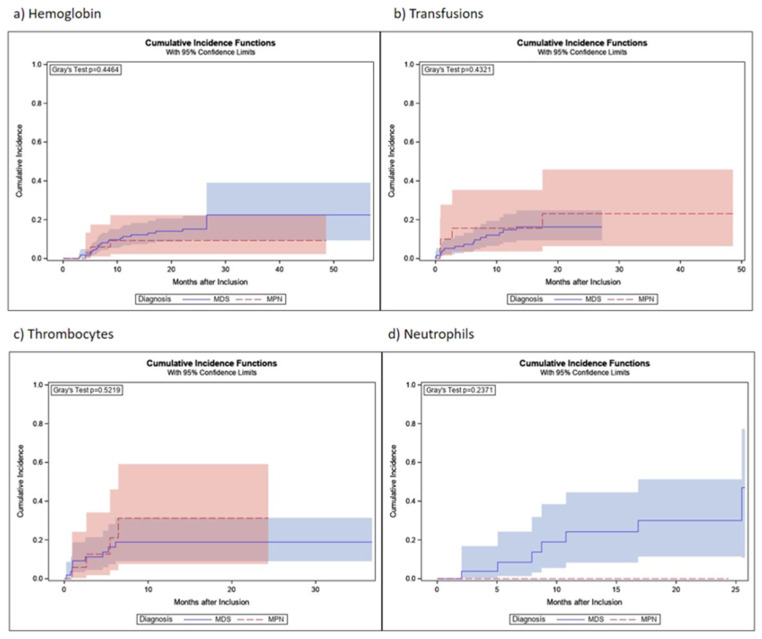
Course time to hematological response (cumulative incidence curves) of hemoglobin, transfusions, platelets, and neutrophils by study months (M1–M24). Colored areas show IQR.

**Table 1 jcm-12-06569-t001:** Patient demographics and disease characteristics.

	n = 403
Sex	
Male	240 (59.6%)
Female	163 (40.4%)
Median age at baseline in years (range)	75 (24–92)
Primary diagnosis	
MDS	246 (61.0%)
MPN	65 (16.1%)
Acute leukemia	20 (5.0%)
Acute myeloid leukemia (AML)	19 (95.0%)
Acute lymphatic leukemia (ALL)	1 (5.0%)
Lymphoma	24 (6.0%)
Multiple myeloma	10 (41.7%)
Non-Hodgkin lymphoma	9 (37.5%)
Chronic lymphatic leukemia (CLL)	5 (20.8%)
Hemoglobinopathies	6 (1.5%)
Thalassemia	5 (83.3%)
Sickle-cell anemia	1 (16.7%)
Anemia, NOS (not MDS-related)	16 (4.0%)
Solid tumor	14 (3.5%)
Condition after stem cell transplantation/(radio-)chemotherapy	6 (1.5%)
Other	6 (1.5%)
Median time from first diagnosis to current iron chelation therapy in months (IQR)	21.6 (8.6–50.0)
Patients with prior iron chelation therapy	21 (5.2%)
Patients with concomitant diseases	364 (90.3%)
Patients with concomitant medication	379 (94.0%)
Transfusions	
Receipt of any transfusions before NIS entry	400 (99.3%)
Median time from primary diagnosis to transfusions in months (IQR)	2.8 (0.2–22.3)
Median time from start of transfusions to start of current iron chelation therapy in months (IQR)	12.1 (5.7–24.3)
Number of erythrocyte concentrates since primary diagnosis until study entry	
0	2 (0.5%)
<20	189 (46.9%)
20–39	109 (27.0%)
40–59	49 (12.2%)
60–79	15 (3.7%)
≥80	22 (5.5%)
Unknown	17 (4.2%)

Abbreviations: IQR: interquartile range, MDS: myelodysplastic syndrome, MPN: myeloproliferative neoplasm, NIS: noninterventional study.

**Table 2 jcm-12-06569-t002:** Specification of MDS/MPN.

**Specification of MDS**	**n = 246**
MDS-SLD	17 (6.9%)
MDS-SLD-RS	30 (12.2%)
MDS-MLD	48 (19.5%)
MDS-MLD-RS	20 (8.1%)
MDS del(5q)	10 (4.1%)
MDS-EB I	32 (13.0%)
MDS-EB II	24 (9.8%)
MDS-U	30 (12.2%)
CMML	9 (3.7%)
MDS/MPN	19 (7.7%)
MDS/MPN-RS	7 (2.8%)
**Specification of MPN**	**n = 65**
PMF	41 (63.1%)
MPN-U	14 (21.5%)
ET	10 (15.4%)

Abbreviations: EB: excess blasts, ET: essential thrombocythemia, MDS: myelodysplastic syndrome, MLD: multi-lineage dysplasia, MPN: myeloproliferative neoplasm, PMF: primary myelofibrosis, RS: ring sideroblasts, SLD: single-lineage dysplasia, U: unclassifiable.

**Table 3 jcm-12-06569-t003:** Treatment changes.

	n = 403
Number of patients with one treatment change	48 (11.9%)
Deferasirox DT to deferasirox FCT	40 (83.3%)
Deferasirox FCT to deferasirox DT	3 (6.3%)
Deferasirox FCT to deferoxamine	3 (6.3%)
Deferoxamine to deferasirox DT	1 (2.1%)
Doxamine to deferasirox FCT	1 (2.1%)
Number of patients with two treatment changes	4 (1.0%)

Abbreviations: DT: dispersible tablet, FCT: film-coated tablet.

**Table 4 jcm-12-06569-t004:** Incidences of AEs/SAEs/SADRs.

	n = 418
Patients with AEs	387 (92.6%)
Patients with non-serious AEs	256 (61.2%)
Patients without AEs	31 (7.4%)
Patients with SAEs	270 (64.6%)
Patients with non-serious ADRs	220 (52.6%)
Patients with SADRs	86 (20.6%)

Abbreviations: AE: adverse event, ADR: adverse drug reaction, SADR: serious adverse drug reaction, SAE: serious adverse event.

**Table 5 jcm-12-06569-t005:** Most common patient-based (S)AEs and (S)ADRs.

	Total	nsAE	SAE	nsADR	SADR
	n = 418	n = 256	n = 270	n = 220	n = 86
General disorders and administration site conditions	202 (48.3%)	109 (42.6%)	82 (30.4%)	53 (24.1%)	11 (12.8%)
General physical health deterioration	51 (12.2%)	17 (6.6%)	25 (9.3%)	7 (3.2%)	3 (3.5%)
Fatigue	41 (9.8%)	31 (12.1%)	2 (0.7%)	8 (3.6%)	1 (1.2%)
Pyrexia	36 (8.6%)	16 (6.3%)	18 (6.7%)	4 (1.8%)	0 (0.0%)
Edema peripheral	23 (5.5%)	17 (6.6%)	2 (0.7%)	4 (1.8%)	0 (0.0%)
Death	19 (4.5%)	0 (0.0%)	16 (5.9%)	0 (0.0%)	3 (3.5%)
Gastrointestinal disorders	187 (44.7%)	80 (31.3%)	44 (16.3%)	113 (51.4%)	16 (18.6%)
Diarrhea	82 (19.6%)	22 (8.6%)	4 (1.5%)	57 (25.9%)	6 (7.0%)
Nausea	50 (12.0%)	26 (10.2%)	3 (1.1%)	22 (10.0%)	3 (3.5%)
Constipation	28 (6.7%)	12 (4.7%)	1 (0.4%)	14 (6.4%)	1 (1.2%)
Infections and infestations	151 (36.1%)	71 (27.7%)	105 (38.9%)	5 (2.3%)	7 (8.1%)
Pneumonia	34 (8.1%)	0 (0.0%)	34 (12.6%)	0 (0.0%)	0 (0.0%)
Urinary tract infection	26 (6.2%)	12 (4.7%)	17 (6.3%)	0 (0.0%)	0 (0.0%)
Nasopharyngitis	23 (5.5%)	21 (8.2%)	2 (0.7%)	0 (0.0%)	0 (0.0%)
Investigations	141 (33.7%)	43 (16.8%)	44 (16.3%)	72 (32.7%)	25 (29.1%)
Blood creatinine increased	41 (9.8%)	7 (2.7%)	0 (0.0%)	29 (13.2%)	7 (8.1%)
Hemoglobin decreased	36 (8.6%)	7 (2.7%)	21 (7.8%)	7 (3.2%)	6 (7.0%)
Serum ferritin increased	17 (4.1%)	2 (0.8%)	0 (0.0%)	14 (6.4%)	1 (1.2%)
Respiratory, thoracic and mediastinal disorders	93 (22.2%)	67 (26.2%)	31 (11.5%)	6 (2.7%)	3 (3.5%)
Dyspnea	32 (7.7%)	25 (9.8%)	7 (2.6%)	1 (0.5%)	0 (0.0%)
Dyspnea exertional	24 (5.7%)	18 (7.0%)	5 (1.9%)	1 (0.5%)	0 (0.0%)
Musculoskeletal and connective tissue disorders	74 (17.7%)	52 (20.3%)	15 (5.6%)	11 (5.0%)	0 (0.0%)
Back pain	15 (3.6%)	13 (5.1%)	2 (0.7%)	0 (0.0%)	0 (0.0%)
Neoplasms benign, malignant and unspecified	73 (17.5%)	1 (0.4%)	68 (25.2%)	0 (0.0%)	6 (7.0%)
Acute myeloid leukemia	24 (5.7%)	0 (0.0%)	23 (8.5%)	0 (0.0%)	1 (1.2%)
Myelodysplastic syndrome	24 (5.7%)	0 (0.0%)	20 (7.4%)	0 (0.0%)	4 (4.7%)
Malignant neoplasm progression	19 (4.5%)	0 (0.0%)	17 (6.3%)	0 (0.0%)	2 (2.3%)
Nervous system disorders	70 (16.7%)	39 (15.2%)	24 (8.9%)	16 (7.3%)	3 (3.5%)
Dizziness	39 (9.3%)	26 (10.2%)	4 (1.5%)	10 (4.5%)	2 (2.3%)
Blood and lymphatic system disorders	66 (15.8%)	15 (5.9%)	43 (15.9%)	6 (2.7%)	12 (14.0%)
Neutropenia	25 (6.0%)	0 (0.0%)	19 (7.0%)	0 (0.0%)	6 (7.0%)
Skin and subcutaneous tissue disorders	60 (14.4%)	30 (11.7%)	6 (2.2%)	22 (10.0%)	7 (8.1%)
Injury, poisoning, and procedural complications	50 (12.0%)	21 (8.2%)	22 (8.1%)	10 (4.5%)	2 (2.3%)
Renal and urinary disorders	49 (11.7%)	9 (3.5%)	20 (7.4%)	8 (3.6%)	15 (17.4%)
Renal failure	14 (3.3%)	0 (0.0%)	4 (1.5%)	0 (0.0%)	10 (11.6%)
Cardiac disorders	45 (10.8%)	3 (1.2%)	41 (15.2%)	3 (1.4%)	3 (3.5%)
Metabolism and nutrition disorders	43 (10.3%)	26 (10.2%)	12 (4.4%)	8 (3.6%)	2 (2.3%)
Vascular disorders	36 (8.6%)	20 (7.8%)	17 (6.3%)	2 (0.9%)	0 (0.0%)
Psychiatric disorders	23 (5.5%)	19 (7.4%)	4 (1.5%)	1 (0.5%)	0 (0.0%)
Hepatobiliary disorders	17 (4.1%)	3 (1.2%)	9 (3.3%)	0 (0.0%)	6 (7.0%)

Abbreviations: AE: adverse event, nsADR: non-serious adverse drug reaction, nsAE: non-serious adverse event, SADR: serious adverse drug reaction, SAE: serious adverse event.

**Table 6 jcm-12-06569-t006:** Hematologic response in patients evaluable for analyses.

	MDS	MPN
Hemoglobin		
Patients analyzed	213	45
Response	21	3
Loss of response	6	0
Death without response	46	11
Transfusions		
Patients analyzed	124	21
Response	15	4
Loss of response	7	1
Death without response	30	5
Platelets		
Patients analyzed	58	18
Response	9	4
Loss of response	3	0
Death without response	18	5
Neutrophils		
Patients analyzed	30	4
Response	7	0
Loss of response	0	0
Death without response	8	1

Abbreviations: MDS: myelodysplastic syndrome, MPN: myeloproliferative neoplasm.

## Data Availability

The datasets used and analyzed during the current study are available from the corresponding author upon reasonable request. The data are not publicly available because of ethical restrictions.

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
