# Peer review of "Iron Chelation in Patients with Myelodysplastic Syndromes and Myeloproliferative Neoplasms—Real-World Data from the German Noninterventional Study EXCALIBUR"

_jcm, 2023, doi:10.3390/jcm12206569_

Round 1

Reviewer 1 Report

The manuscript investigates the effect of iron chelation in MDS and MPS patients. The authors are to be congratulated for this huge effort. This reviewer has minor comments only:

- Are differences observed between MDS and MPD patients ? this could be discussed in some more details.

- In line with this: are there subgroups benefitting more than others ?

- if so, are thse subgroups charaterized by genetic disease-related abnormalities ?

- can the authors speculate on desitable ferritin levels as a therapeutic goal ?

- what is the trigger to initiate iron chelation? a specific ferritin evel? at the discretion of the treating physician? there may be a huge variation between centers, this bias could be discussed.

Reviewer 2 Report

A few issues:

The graphics in the manuscript are relatively unattractive and low-quality, and could be made more hlepful for the readers.

I am concerned with the very large drop-out of patients during control visits (M9, M12, ...). Could the group of patients who stay in observation by biased by their blood counts, transfusion, satisfaction, etc.?

With respect to 'hematological response' -the impact and proportions definitely depends on the composition of MDS subtypes and also on treatment? Also it changes over decades when compared to previous cohorts. 

What about reponse analysis straftified by doseases /interventions?

Lastly, the fact that patients with lower ferritin fare better is a tautology. If they have less transfusions they have milder disease and fare better to begin with.

Round 2

Reviewer 2 Report

I ackowledge the responses provided by the authors. Just the Figures 1 and 3 are there difficult to read, they need improving.